# The Incidence of Myocarditis and Pericarditis in Post COVID-19 Unvaccinated Patients—A Large Population-Based Study

**DOI:** 10.3390/jcm11082219

**Published:** 2022-04-15

**Authors:** Ortal Tuvali, Sagi Tshori, Estela Derazne, Rebecca Regina Hannuna, Arnon Afek, Dan Haberman, Gal Sella, Jacob George

**Affiliations:** 1Heart Center, Kaplan Medical Center, Rehovot, Hebrew University of Jerusalem, Jerusalem 91905, Israel; ortalhaga@clalit.org.il (O.T.); danha1@clalit.org.il (D.H.); galse1@clalit.org.il (G.S.); 2Research Authority, Rehovot, Hebrew University of Jerusalem, Jerusalem 91905, Israel; sagitz1@clalit.org.il (S.T.); rebeccar@clalit.org.il (R.R.H.); 3Sackler Faculty of Medicine, Tel Aviv University, Tel Aviv 6997801, Israel; estela.simhoni@gmail.com (E.D.); arnon.afek@sheba.health.gov.il (A.A.); 4General Management, The Chaim Sheba Medical Centre, Tel Hashomer, Ramat-Gan 52621, Israel

**Keywords:** COVID-19, myocarditis, pericarditis

## Abstract

Myocarditis and pericarditis are potential post-acute cardiac sequelae of COVID-19 infection, arising from adaptive immune responses. We aimed to study the incidence of post-acute COVID-19 myocarditis and pericarditis. Retrospective cohort study of 196,992 adults after COVID-19 infection in Clalit Health Services members in Israel between March 2020 and January 2021. Inpatient myocarditis and pericarditis diagnoses were retrieved from day 10 after positive PCR. Follow-up was censored on 28 February 2021, with minimum observation of 18 days. The control cohort of 590,976 adults with at least one negative PCR and no positive PCR were age- and sex-matched. Since the Israeli vaccination program was initiated on 20 December 2020, the time-period matching of the control cohort was calculated backward from 15 December 2020. Nine post-COVID-19 patients developed myocarditis (0.0046%), and eleven patients were diagnosed with pericarditis (0.0056%). In the control cohort, 27 patients had myocarditis (0.0046%) and 52 had pericarditis (0.0088%). Age (adjusted hazard ratio [aHR] 0.96, 95% confidence interval [CI]; 0.93 to 1.00) and male sex (aHR 4.42; 95% CI, 1.64 to 11.96) were associated with myocarditis. Male sex (aHR 1.93; 95% CI 1.09 to 3.41) and peripheral vascular disease (aHR 4.20; 95% CI 1.50 to 11.72) were associated with pericarditis. Post COVID-19 infection was not associated with either myocarditis (aHR 1.08; 95% CI 0.45 to 2.56) or pericarditis (aHR 0.53; 95% CI 0.25 to 1.13). We did not observe an increased incidence of neither pericarditis nor myocarditis in adult patients recovering from COVID-19 infection.

## 1. Introduction

Coronavirus disease 2019 (COVID-19), caused by severe acute respiratory syndrome coronavirus 2 (SARS-CoV-2), is a leading cause of morbidity and mortality worldwide [1]. In addition to the clinical manifestations during the acute phase of the COVID-19 disease, there is an accumulating data regarding the subacute and long-term effects of COVID-19, also known as “post-acute COVID-19 syndrome” or “Long COVID”, defined by persistent symptoms several weeks after onset of COVID-19 infection [2]. The “Long-COVID” or “post-acute COVID-19 syndrome” is characterized by multi-organ sequelae or persistent symptoms after recovering from the acute COVID-19 phase, generally after 3 to 4 weeks from the onset of symptoms or the first PCR positive result test [3].

The pathogenesis of “Long-COVID” may result from several mechanisms, including direct viral toxicity, hypercoagulability, microvascular injury, and angiotensin-converting enzyme maladaptation [4]. While the underlying pathophysiological mechanisms leading to post-acute COVID-19 are yet to be fully understood, immune-mediated response [5,6] and immune dysregulation [4] are believed to play a major contributing role in the pathogenesis of this syndrome. There is no consensus as to which time point represents the transition from the acute COVID-10 infection and the recovery phase. As we were considering an indirect immune-mediated inflammation as the potential mechanism explaining delayed peri/myocarditis we reasoned that 10 days after infection is a relevant time point as this is valid with regard to pericarditis after myocardial infarction (Dressler syndrome) or cardiac surgery (postpericardiotomy syndrome).

Infectious causes have been shown to be an important inciting event in the pathophysiology of autoimmune diseases [7]. Viral infections have also been associated with the presence of autoimmune diseases such as systemic lupus disease, rheumatoid arthritis, and diabetes mellitus [8].

In viral infections, both direct contact and replication-induced injury and autoimmune mechanisms have been implicated in the pathogenesis of consequent myocarditis. Inappropriate regulation of T cells resulting from pathogen-related altered self -proteins or from molecular mimicry between the virus and the host or over-activation of B cells may lead to inappropriate immune-mediated damage to host tissue [9]. As such, this delayed inappropriate response may trigger autoimmune like myocarditis and pericarditis [10]. Accordingly, in a subset of patients with idiopathic dilated cardiomyopathy, prior myocarditis is considered causative in some cases and an example supportive of associating etiopathogenesis is the occurrence of anti-beta 1 adrenoreceptor antibodies [11].

Several autoimmune phenomena were linked to a previous COVID-19 infection including heparin-induced thrombocytopenia (HITT), Kawasaki-like syndromes (MIS-C and MIS-A), Guillain-Barre syndrome, vasculitis, and thyroiditis [8]. Thus, it can be postulated that the risk for autoimmune induced myocarditis and pericarditis is increased in recovering COVID-19 patients.

It has recently been reported that the incidence of myocarditis and pericarditis is increased in COVID-19 patients during the acute illness [12]. However; whether or not myocarditis and pericarditis after the recovery period are a part of the long COVID-19 syndrome is yet unknown. Herein, we studied the incidence of myocarditis and pericarditis in a large cohort of COVID-19 patients after recovering from the acute infection.

## 2. Methods

### 2.1. Study Setting

We retrieved observational data from Clalit Health Services (CHS). CHS is the largest of four health maintenance organizations that offer mandatory health care coverage in Israel. CHS insures over 50% of the Israeli population (>4.4 million persons), and the CHS-insured population is approximately representative of the Israeli population at large [12,13]. CHS provides outpatient care, and inpatient care is divided between CHS and other hospitals. Research data can be retrieved from the central data warehouse using the CHS Secure Data Sharing Platform powered by MDClone (https://www.mdclone.com, accessed on 5 September 2021).

This study was approved by the institutional review board, and was exempt from the requirement for informed consent.

### 2.2. Study Design and Patient Population

We retrieved records of all adult patients (age ≥ 18 years) who had a documented positive COVID-19 PCR test (*n* = 213,624) between 7 March 2020 and 31 January 2021 (Figure 1A). Records included demography and cardiovascular risk factors: smoking status, obesity, diabetes mellitus, hyperlipidemia, CKD (chronic kidney disease), PVD (peripheral vascular disease, ACS (acute coronary syndrome), essential hypertension, CVA (cerebrovascular accident), and heart failure (for the The International Classification of disease (ICD-10) code list for all diagnoses, see Appendix A). Diagnostic inpatient codes for myocarditis (I40, I40.9, I51.4) and pericarditis (I30, I30.0, I30.9) were extracted between 10 days after COVID-19 infection and earliest between 6 months from infection or 28 February 2021. This was done to ensure a minimum follow-up period of 18 days and a maximum follow-up period of six months. Since data was retrieved during October 2021, we allowed for over 6 months delay in data transfer between hospitals that do not belong to the CHS and the CHS data warehouse. The post-COVID timeframe was defined from at least ten days after the date of positive PCR test contingent upon lack of symptoms related to COVID-19 infection, according to the definitions of the Israeli Ministry of Health. Patients with a first vaccination received before COVID-19 infection were excluded (*n* = 16,632), resulting in the final COVID-19 cohort (*n* = 196,992).

A control group was created from a cohort of adult patients with at least one negative COVID-19 PCR between 7 March 2020 and 15 December 2020, and with no prior positive COVID-19 PCR before retrieval of data in August 2021 (*n* = 935,976). 15 December 2020 was selected as the stop date, since the massive Israeli vaccination campaign was initiated on 20 December 2020. Still, five patients were excluded due to a previous COVID-19 vaccination. From this pool of patients (*n* = 935,971), the control cohort was created by 3:1 matching of age (±2 years) and sex (*n* = 590,976). The follow-up period of each of the three control patients was set to the exact same length of follow-up of the matched COVID-19 patient. The follow-up period was calculated backwards from 15 December 2020 (Figure 1B), in order to avoid the potential impact of COVID-19 vaccination on myocarditis and pericarditis. Although the COVID-19 patients’ drafting period was from March 2020 to January 2021, almost all COVID-19 cases occurred between July 2020 and January 2021 (Appendix A).

### 2.3. Statistical Analysis

We compared the baseline characteristics of the cohorts with the chi-square test. We used Kaplan-Meier cumulative incidence curves to assess the effect of post-COVID-19 infection on myocarditis and pericarditis up to a maximal follow-up of six months. Univariable and multivariable Cox proportional hazards regression models were used. Post-COVID infection, age, sex, BMI, diabetes, hyperlipidemia, obesity, chronic kidney injury, smoking status, peripheral vascular disease, acute coronary syndrome, and essential hypertension were introduced in the adjusted models. Both crude hazard ratio (HR) and adjusted HR (aHR) are presented with 95% confidence intervals. A *p*-value less than 0.05 was considered statistically significant. Statistical analyses were performed with R version 4.0.2, packages: survival, Survminer, ggplot2, and with SPSS version 26 (IBM).

## 3. Results

A total of 787,968 Clalit Health Services adult members (age ≥ 18) were included in the study (Figure 1), comprising of COVID-19 cohort (*n* = 196,992) and a sex and age 3:1 matched control cohort (*n* = 590,976). Total follow-up was 700,040 person-months in the COVID-19 cohort, and 2,100,077 person-months in the control cohort, with a median follow-up of 4.1 months [IQR 1.3–5.6 months]. The mean standard deviation (SD) age in both groups was 42.4 (17.7) years, and 45.7% were males (Table 1). There was a slightly higher BMI (mean (SD) 27.1 (7.1) vs. 26.1 (6.7) kg/m^2^) with higher prevalence of obesity (29.9% vs. 24.6%), diabetes mellitus (13% vs. 10.5%), essential hypertension (18.5% vs. 17%), cerebrovascular accidents (4% vs. 3.7%) and heart failure (2.6% vs. 2.2%) in the COVID-19 cohort. There was a lower prevalence of current and past smoking (11.3% vs. 18.4% and 10.8% vs. 12.3%) and peripheral vascular disease (1.2% vs. 1.4%) in the COVID-19 cohort.

During the study period, nine cases of myocarditis and 11 cases of pericarditis were detected in the COVID-19 cohort (Table 2). Twenty-seven cases of myocarditis and 52 cases of pericarditis were detected in the control cohort. Two out of the nine myocarditis patients were hospitalized due to severe COVID-19 infection with the need for mechanical ventilation, and myocarditis was diagnosed during the COVID-19 hospitalization at days 19 and 37 after infection, respectively. None of the patients who were diagnosed with pericarditis were hospitalized due to COVID-19 infection. The median (IQR) duration of hospitalization following myocarditis in the COVID-19 cohort was 5 (2–25) days vs. 3 (3–5) days in the control cohort (*p* = 0.291). The median (IQR) duration of hospitalization following pericarditis was 2 (2–3.5) days in the COVID-19 cohort and 3 (1.8–5.3) days in the control cohort (*p* = 0.272).

No statistical difference in the incidence rate of both myocarditis (*p* =1) and pericarditis (*p* =0.17) was observed between the COVID-19 cohort and the control cohort (Figure 2).

In the multivariable Cox proportional hazards regression model (Table 3, Appendix A), age (aHR 0.96; 95% CI 0.93 to 1.00, *p* =0.045) and the male sex (aHR 4.42; 95% CI 1.64 to 11.96, *p* = 0.003) were independently associated with myocarditis. Obesity was borderline associated with myocarditis (aHR 2.31; 95% CI 0.99 to 5.41, *p* = 0.053). Post COVID-19 infection was not associated with myocarditis (aHR 1.08; 95% CI 0.45 to 2.56, *p* = 0.869).

Male sex (aHR 1.93; 95% CI 1.09 to 3.41, *p* = 0.025) and peripheral vascular disease (aHR 4.20; 95% CI 1.50 to 11.72, *p* = 0.006) were associated with pericarditis (Table 3). Post COVID-19 infection was not associated with pericarditis (aHR 0.53; 95% CI 0.25 to 1.13, *p* = 0.1).

## 4. Discussion

In the current large population study of subjects, who were not vaccinated against SARS-CoV-2, we observed no increase in the incidence of myocarditis or pericarditis from day 10 after positive SARS-CoV-2.

Multivariable analysis did show male sex as associated with a higher risk of developing myocarditis or pericarditis, regardless of previous COVID-19 infection.

COVID-19 infection is responsible for considerable morbidity and mortality at an unprecedented scale globally. Cumulative scientific and clinical data is evolving on the sub-acute and long-term effects of COVID-19 infection, which can affect multiple organ systems. Recent studies suggest several mechanisms for the pathogenesis of the persistent and prolonged signs and symptoms associated with the cardiovascular system.

Several earlier studies on SARS-CoV infections have highlighted the possible link between types of coronavirus infections and immune-mediated responses [14,15].

Antibodies to Coronavirus-OC43 and 229E were found in patients diagnosed with multiple sclerosis [16]. High and sustained levels of anti-erythrocyte autoantibodies were found in mice infected with murine hepatitis Coronavirus [17].

Given the clinical similarities between SARS-CoV-2 and other coronaviruses [18,19], it is conceivable that immune-mediated tissue damage is also potentially induced by SARS-CoV-2 infection. Molecular mimicry, activation of specific T cells to virus altered self-proteins, and activation of B cells constitute a few examples of mechanisms mediating the potential occurrence of myocarditis and pericarditis that could play a role in triggering delayed cardiac inflammation [9]. Furthermore, in a recent study 309 COVID-19 patients were tested for the presence of six different autoantibodies, such as anti-nuclear antibodies and anti-Interferon-α2 antibodies, at least two months after the initial illness. In this study, autoantibodies were associated with a higher risk of developing post-acute sequelae of COVID-19 [20].

Early in the COVID-19 pandemic, it was evident that COVID-19 patients with cardiovascular comorbidities have worse prognosis and higher in-hospital mortality [21]. Individuals with underlying autoimmune diseases appear to be particularly vulnerable to severe sequelae resulting from COVID-19 infection [22]. Other studies demonstrated that severe COVID-19 disease is associated with robust inflammatory responses including type two and hyper four hypersensitivity responses, resulting from overactivation of T cells and a subsequent cytokine storm [23,24]. Immune-mediated manifestations of COVID-19 include mimicry of autoimmune diseases like Kawasaki disease, Guillain-Barre syndrome, vasculitis, myositis, and myocardial damage [25].

Puntmann et al. found a 78% cardiac involvement assessed by Cardiac Magnetic Resonance Imaging (MRI) among patients with a confirmed diagnosis of COVID-19 eight weeks before enrollment [26], most of whom were asymptomatic or had just mild symptoms. This study demonstrates cardiac inflammation independent of the severity of the initial illness nor the overall course of the acute illness. A recent study showed an increased risk of late cardiovascular outcomes in either symptomatic or asymptomatic SARS-CoV-2 infection [27]. Thus, we also sought to investigate whether cardiac damage may also occur regardless of the presence of corona-related symptoms. Further corroborating the potential late inflammatory pericardial and myocardial involvement is an additional cardiovascular magnetic resonance (CMR) guided study demonstrating myocardial edema, fibrosis, and impaired right ventricle function in 58% (16 of 25) patients recently recovering from COVID-19 [28]. These studies are supported by an autopsy study pointing to the occurrence of mononuclear infiltrates in patients with COVID-19 infection [29]. However, an important caveat to these studies is the lack of appropriate controls including patients with other intercurrent viral infections.

Similar to our study, Xie et al. showed that individuals with COVID-19 infection are at increased risk of cardiovascular complications 30 days after infection, including pericarditis and myocarditis regardless of the need for hospitalization [30]. Comparable with our study, the study population was tested for the risk of inflammatory heart diseases regardless of previous SRAS-COV-2 vaccination. Yet, in contrast, in the study by Xie et al., the tested cohort was homogenous, comprising of US Department of Veterans Affairs with male predominance and young age. The difference in the population characteristics may explain the dissimilarity between the results of the studies as young males are known to exhibit a higher incidence of myocarditis and pericarditis.

Higher risk of myocarditis (risk ratio, 18.28; 95% CI, 3.95 to 25.12; risk difference, 11.0 events per 100,000 persons; 95% CI, 5.6 to 15.8) and pericarditis was observed in a large population study of recently published by Barda et al. [12]. Although both our study and the study by Barda et al. are based on Clalit Health Service patients, there are several important differences between the studies. Barda et al. were focused on COVID-19 vaccination, and thus the matching was designed to neutralize vaccination-related factors, while our study is on a non-vaccinated population. Barda et al. studied the occurrence of myocarditis and pericarditis from positive PCR results up to 42 days, while we study recovering patients starting 10 days after infection and for a significantly more prolonged time. Barda et al.’s analysis also ignores the timing of myocarditis and pericarditis. Finally, while Barda et al. have included many causes of myocarditis and pericarditis, we only included acute myocarditis and pericarditis in hospitalized patients which is more likely to be accurate.

Our current study has several limitations. First, although the potential number of participants who were considered for inclusion was large, the number of cases of myocarditis and pericarditis was small. This was mainly attributed to the limitation of a relatively short follow-up period due to the initiation of the massive vaccination program. Second, we included only cases of hospitalized myocarditis or pericarditis patients, whereas outpatient medical records were excluded from the study. This could possibly omit a small number of patients with mild disease. Furthermore, we included a diagnosis of myocarditis and pericarditis according to the medical records, without access to patient-based information regarding confirmation of the diagnosis.

## 5. Conclusions

Our data suggest that there is no increase in the incidence of myocarditis and pericarditis in COVID-19 recovered patients compared to uninfected matched controls. Further longer-term studies will be needed to estimate the incidence of pericarditis and myocarditis in patients diagnosed with COVID-19.

## Figures and Tables

**Figure 1 jcm-11-02219-f001:**
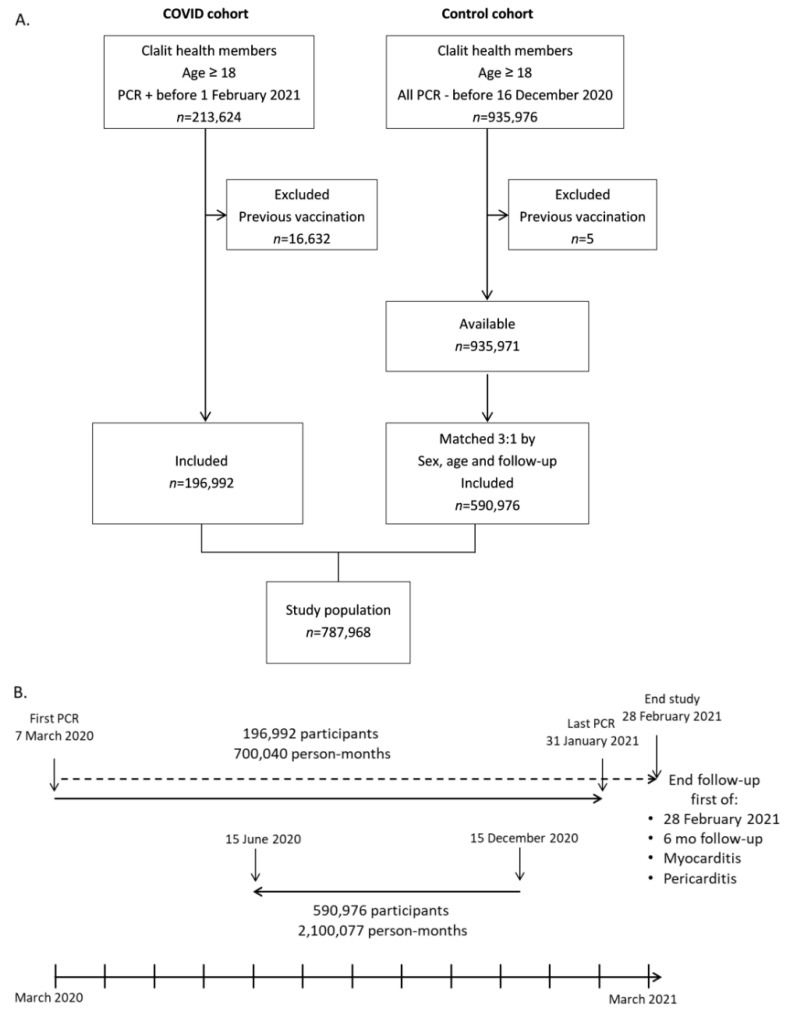
Study profile, design and timelines. ‘clalit’ = Clalit Health Services. Vaccination = COVID-19 vaccination. PCR −/+ = negative/positive COVID-19 PCR. (**A**). Flowchart of patient selection in COVID and control cohorts. (**B**). Research timeline from first PCR in 7 March 2020 to the end of study (lower part). The COVID cohort is depicted in the upper part, and the matching period of the control cohort is depicted in the middle.

**Figure 2 jcm-11-02219-f002:**
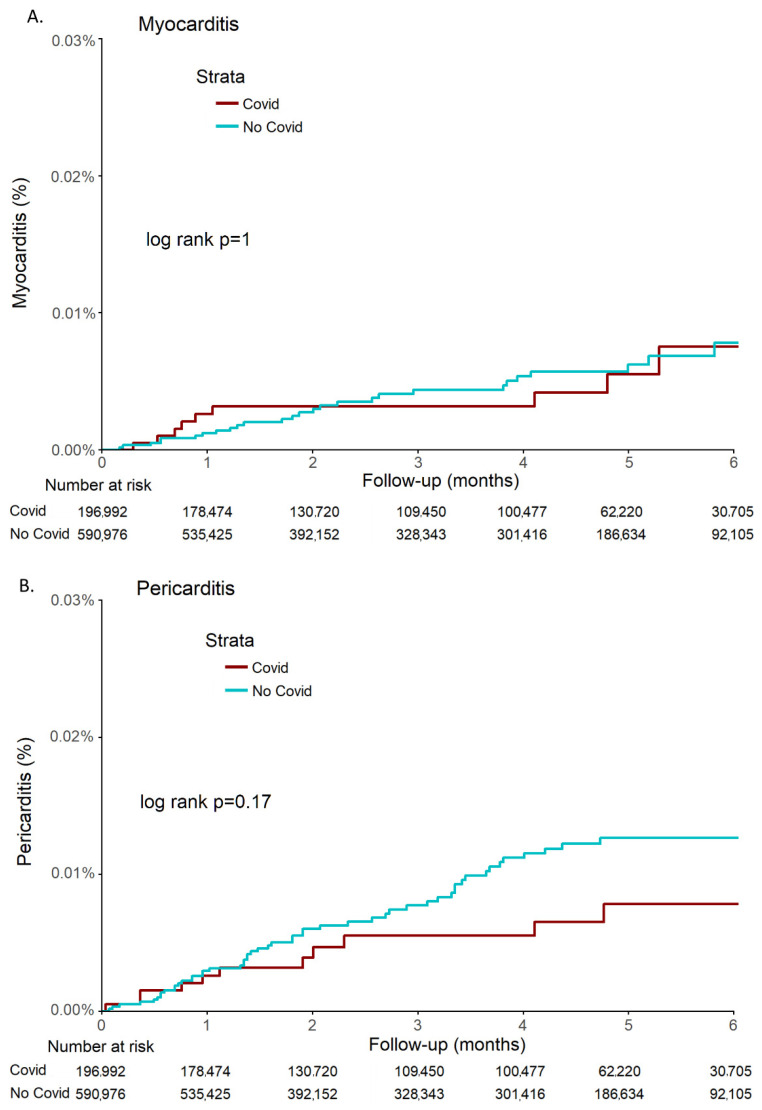
Kaplan-Meier estimates of cumulative probability of myocarditis (**A**) and pericarditis (**B**) in COVID-19 and control cohorts during 6 months.

**Table 1 jcm-11-02219-t001:** Baseline Characteristics of the study population by COVID-19 infection status.

	Control	COVID-19
	(*n* = 590,976)	(*n* = 196,992)
Age (year)	42.4 (17.7)	42.4 (17.7)
Sex		
Male	270,210 (45.7)	90,070 (45.7)
Female	320,766 (54.3)	106,922 (54.3)
BMI (kg/m^2)^	26.1 (7.1)	27.1 (6.7)
Sector		
Arab	109,759 (18.6)	58,841 (29.9)
Bedouin	19,956 (3.4)	5306 (2.7)
Jewish	436,986 (74.1)	108,360 (55.0)
Other	23,285 (3.9)	24,475 (12.4)
Smoking		
Never	383,002 (69.3)	137,079 (78.0)
Current	101,475 (18.4)	19,824 (11.3)
Past	67,992 (12.3)	18,917 (10.8)
Obesity	145,537 (24.6)	58,807 (29.9)
Diabetes	61,978 (10.5)	25,583 (13.0)
Hyperlipidemia	183,704 (31.1)	60,629 (30.8)
CKD	18,593 (3.1)	6725 (3.4)
PVD	8016 (1.4)	2448 (1.2)
Hypertension	100,736 (17.0)	36,502 (18.5)
ACS	32,987 (5.6)	11,027 (5.6)
CVA	22,032 (3.7)	7833 (4.0)
Heart failure	13,228 (2.2)	5132 (2.6)

Variables were expressed as no. (%) or as mean (SD). CKD = chronic kidney disease. PVD = peripheral vascular disease, CVA = Cerebrovascular Accident. ACS = acute coronary syndrome.

**Table 2 jcm-11-02219-t002:** Baseline characteristics of patients with myocarditis and pericarditis patients in the COVID and no COVID cohorts.

		Myocarditis			Pericarditis		
		No COVID (*n* = 27)	COVID (*n* = 9)	*p*-Value	No COVID (*n* = 52)	COVID (*n* = 11)	*p*-Value
Days in hospital		3 [3–5]	5 [2–25]	0.291	3 [1.8–5.3]	2 [2–3.5]	0.272
Age (year)		39.1 (16.8)	36.4 (19.7)	0.695	49.1 (20.3)	45.6 (19.3)	0.609
Sex (male)		20 (74.1)	8 (88.9)	0.643	31 (59.6)	8 (72.7)	0.637
BMI (kg/m^2^)		27.3 (6.1)	26.3 (6.9)	0.713	26.5 (5.8)	28.6 (3.8)	0.319
Diabetes		4 (14.8)	1 (11.1)	1.000	12 (23.1)	1 (9.1)	0.528
Hyperlipidemia		6 (22.2)	2 (22.2)	1.000	25 (48.1)	4 (36.4)	0.708
Obesity		14 (51.9)	1 (11.1)	0.079	15 (28.8)	6 (54.5)	0.197
CKD		3 (11.1)	1 (11.1)	1.000	7 (13.5)	1 (9.1)	1.000
Smoking	Now	6 (26.1)	1 (12.5)	0.602	9 (19.1)	1 (9.1)	0.777
	Past	3 (13.0)	2 (25.0)		7 (14.9)	1 (9.1)	
PVD		2 (7.4)	0 (0.0)	1.000	4 (7.7)	2 (18.2)	0.609
ACS		3 (11.1)	2 (22.2)	0.781	9 (17.3)	2 (18.2)	1.000
Hypertension		5 (18.5)	2 (22.2)	1.000	15 (28.8)	3 (27.3)	1.000
CVA		3 (11.1)	2 (22.2)	0.781	5 (9.6)	2 (18.2)	0.769
HF		5 (18.5)	2 (22.2)	1.000	6 (11.5)	1 (9.1)	1.000
Severe COVID		-	2 (22.2)	NA	-	0 (0.0)	NA

Variables were expressed as no. (%), median [IQR], or as mean (SD). CKD = chronic kidney disease. PVD = peripheral vascular disease. CVA = Cerebrovascular Accident, ACS = acute coronary syndrome. HF = Heart Failure. Days in hospital = length of stay of the hospitalization at which myocarditis/pericarditis were diagnosed.

**Table 3 jcm-11-02219-t003:** Adjusted HRs for myocarditis and pericarditis.

	Myocarditis		Pericarditis	
	aHR (95% CI)	*p*-Value	aHR (95% CI)	*p*-Value
COVID-19	1.08 (0.45–2.56)	0.869	0.53 (0.25–1.13)	0.100
Age	**0.96 (0.93–1.00)**	**0.045 ***	1.01 (0.99–1.03)	0.537
Sex (male)	**4.42 (1.64–11.96)**	**0.003 ***	**1.93 (1.09–3.41)**	**0.025 ***
BMI	1.00 (0.97–1.04)	0.935	1.00 (0.94–1.06)	0.897
Diabetes	1.15 (0.26–5.00)	0.856	0.97 (0.43–2.21)	0.950
Hyperlipidemia	0.33 (0.08–1.43)	0.139	1.13 (0.54–2.38)	0.749
Obesity	2.31 (0.99–5.41)	0.053	1.32 (0.64–2.71)	0.448
CKD	3.80 (0.82–17.66)	0.088	1.91 (0.72–5.05)	0.191
Smoking (Now)	1.62 (0.65–4.06)	0.304	0.85 (0.42–1.74)	0.661
Smoking (Past)	1.85 (0.61–5.64)	0.277	0.73 (0.33–1.63)	0.439
PVD	1.35 (0.14–12.84)	0.793	**4.20 (1.50–11.72)**	**0.006 ***
ACS	3.93 (0.76–20.40)	0.104	1.52 (0.61–3.80)	0.366
Hypertension	1.46 (0.36–5.87)	0.592	0.88 (0.38–2.06)	0.770

CKD = chronic kidney disease. PVD = peripheral vascular disease. ACS = acute coronary syndrome. * Statistically significant results are highlighted in bold.

## Data Availability

The data presented in this study is available on request from the corresponding author.

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
