# Peer review of "The Incidence of Myocarditis and Pericarditis in Post COVID-19 Unvaccinated Patients—A Large Population-Based Study"

_jcm, 2022, doi:10.3390/jcm11082219_

Round 1

Reviewer 1 Report

The authors present very interesting study based on a big database from insurance service aiming to assess the incidence of post-acute COVID-19 myocarditis and pericarditis in non-vaccinated patients with PCR + test. They compared the incidence of myocarditis and pericarditis  in COVID-19 patients to the incidence in non-vaccinated subjects with negative PCR test and no positive PCR test. In the large population cohort the incidence of myocarditis and pericarditis was extremely low and there was no increased incidence associated with COVID-19.

My congratulations to the authors for the excellent work! I do not have comments. 

Author Response

Thank you for your kind reply and consideration, I highly appreciate it. 

Reviewer 2 Report

The submitted study sought to investigate the incidence of myocarditis and pericarditis in unvaccinated Israeli population relative to COVID-19 positive peers in the acute-post infectious stage (10days-6months). The article is well written, easy to follow, but would benefit from some major and minor improvements for readability and data clarity.

Major Points:

Title: Recommend highlighting the uniqueness of the paper by including “unvaccinated status” within the title.

Suggestion: Myocarditis and Pericarditis Incidence in Post COVID-19 Positive Unvaccinated Patients – A Large Population-Based Study

Abstract: Currently 252 words, ideally closer to 200. Journal recommends not using titles for each section (Background, Methods, etc.). Would include number of control patients (590,976) to support title and that CHS is in Israel.

Line 97: Why not include codes I40.0 (infective myocarditis) and I30.1 (infective pericarditis)? Your assumption is that the COVID-19 is the direct cause of the cardiac inflammation for the positive cohort.

Line: 289: The current submission is missing a lot of the “Back Matter” – Author Contributions, Funding, Institutional Review Board Statement, Informed Consent Statement, Data Availability Statement, Conflicts of Interest. Though not required at initial submission, it does make a complete review difficult.

Minor Points:

Need to define abbreviations upon first use throughout the article: ICM-10, CI, CMR, etc.

Line 4: The symbols for equal participation as authors is incorrect, the corresponding author symbol is used and simultaneously does not appear for Prof. George.

Line 107: The “e” designation preceding Table or Figure is not consistent with the format of this journal, recommend eliminating and using plain numbers (Figure 1, Figure 2, etc.). If the authors intent is to include these figures/tables as supplemental information, please adjust as Figure S1, Table S1, etc. per the journal template.

Line 96: Table S1 in text appears to be eTable1 (see earlier comment to comply with journal template and be consistent style throughout the article).

Line 128: Statistical Analysis – What p-value is considered significant? p<0.05?

Line 183: Would be helpful if statistically significant values would be identified (bold or symbols) to increase readability for Table 3, eTable 2, and eTable 3.

Line 208: citation numbers appear to not be uniform in font/size to other citations.

Line 231+235+246: citation should follow comma or period (,26 / .27 / .30)

Line 248: period, not comma after “vaccination”

Line 272: add period after “mild disease”

Author Response

Dear reviewer, thank you very much for reviewing our manuscript, your comments are much appreciated.  Find enclosed our comment regarding your review:

Point 1- Title: Recommend highlighting the uniqueness of the paper by including “unvaccinated status” within the title.

Response 1: The title of the article was changed according to your suggestion

Point 2- Abstract: Currently 252 words, ideally closer to 200. Journal recommends not using titles for each section (Background, Methods, etc.). Would include number of control patients (590,976) to support title and that CHS is in Israel.

Response 2: We shortened the abstract to 236 words (after adding the cohort sample size and the location of CHS. we are afraid that omitting more words from the current abstract will harm the main message of the article.

Point 3: Why not include codes I40.0 (infective myocarditis) and I30.1 (infective pericarditis)? Your assumption is that the COVID-19 is the direct cause of the cardiac inflammation for the positive cohort.

Response 3:  Much obliged. During Data retrieval from our internal institutional coding system, we have selected broad definitions that include additional specific definitions of the main outcomes. These codes are mentioned in the manuscript.

In addition, we decided to use the most frequent definitions of pericarditis and myocarditis used in our coding system.

Furthermore, in order to specify our results as for acute pericarditis and myocarditis only, we decided to omit the definition of infective pericarditis and infective myocarditis due to the fact that these definitions include bacterial infections, parasitic infections and recurrent infections as well.

Point 4: The current submission is missing a lot of the “Back Matter” – Author Contributions, Funding, Institutional Review Board Statement, Informed Consent Statement, Data Availability Statement, Conflicts of Interest. Though not required at initial submission, it does make a complete review difficult.

Response 4: In the current corrected submission, we added the following information- Author Contributions, Funding, Institutional Review Board Statement, Informed Consent Statement, Data Availability Statement, Conflicts of Interest (page-14).

Author Contributions: Conceptualization- J.G; Data curation- E.D; Formal analysis- E.D and R.R.H; Investigation, S.T, O.T and D.H; Methodology – E.D and G.S; Project administration, J.G; Supervision, A.A; Validation, S.T and J.G; Writing – original draft, O.T; Writing – review & editing, S.T and J.G. All authors have read and agreed to the published version of the manuscript

Funding: This research received no external funding.

Institutional Review Board Statement: The study was conducted in accordance with the Declaration of Helsinki, and approved by the Institutional Review Board of Kaplan Medical Center (KMC-123-2021), Date of approval 06/24/2021.

Informed Consent Statement: Patient consent was waived due to de-identified retrospective cohort study.

Acknowledgments: The authors would like to thank the Kaplan Medical Center Heart team for their support.

Data Availability Statement: The data presented in this study is available on request from the corresponding author.

Conflicts of Interest: The authors declare no conflict of interest.

Point 5: Need to define abbreviations upon first use throughout the article: ICM-10, CI, CMR, etc.

Response 5:  We defined our abbreviations according to your suggestion. I added the following abbreviations: CMR, MRI, CVA, SD,  US and CI. 

Point 6: The symbols for equal participation as authors is incorrect, the corresponding author symbol is used and simultaneously does not appear for Prof. George.

Response 6: Dr. Ortal Tuvali and Dr. Sagi Tshori were equally contributed to this work. We changed the symbols according to the format of this journal.

Point 7+8: The “e” designation preceding Table or Figure is not consistent with the format of this journal, recommend eliminating and using plain numbers (Figure 1, Figure 2, etc.). If the authors intent is to include these figures/tables as supplemental information, please adjust as Figure S1, Table S1, etc. per the journal template.

Table S1 in text appears to be eTable1 (see earlier comment to comply with journal template and be consistent style throughout the article).

Response 7+8: We changed the numbers of the following figures and tables: eFigure 1 was changed to Figure S1. eTable 2 was changed to table S2, eTable 3 was changed to Table S3.

Point 9: Statistical Analysis – What p-value is considered significant? p<0.05?

Response 9: Yes. We defined P value less than 0.05 as statistically significant. This note was added to the methods description.

Point 10: Would be helpful if statistically significant values would be identified (bold or symbols) to increase readability for Table 3, eTable 2, and eTable 3.

Response 10: We have highlighted the statistically significant values according to your note.

Point 11: citation numbers appear to not be uniform in font/size to other citations.

Response 11: Corrected

Point 12: citation should follow comma or period (,26 / .27 / .30)

Answer 12: Corrected

Point 13 period, not comma after “vaccination”

Answer 13: Corrected

Point 14 add period after “mild disease”

Answer 14: Corrected

Thank you in advance,

Dr. Ortal Tuvali.

Reviewer 3 Report

The article brings very important data originating from a large population-based study. Overall, studies are rigorously performed, and the data are convincing. The study limitation section has answered the queries that I had while reading the manuscript. Hence no major comments.

Author Response

Dear Editor, Much appreciated your review regarding our manuscript.